# A WoT-Based Method for Creating Digital Sentinel Twins of IoT Devices

**DOI:** 10.3390/s21165531

**Published:** 2021-08-17

**Authors:** Ivan Lopez-Arevalo, Jose Luis Gonzalez-Compean, Mariana Hinojosa-Tijerina, Cristhian Martinez-Rendon, Raffaele Montella, Jose L. Martinez-Rodriguez

**Affiliations:** 1Cinvestav Tamaulipas, Victoria 87130, Mexico; ilopez@cinvestav.mx (I.L.-A.); mariana.hinojosa@cinvestav.mx (M.H.-T.); 2ARCOS Research Group, University Carlos III of Madrid, 28911 Leganes, Spain; cristhma@pa.uc3m.es; 3Department of Science and Technologies, University of Napoli Parthenope, 80133 Napoli, Italy; raffaele.montella@uniparthenope.it; 4Reynosa Rodhe Multidisciplinary Academic Unit, Autonomous University of Tamaulipas, Reynosa 88779, Mexico; lazaro.martinez@uat.edu.mx

**Keywords:** digital twins, IoT data, microservices, cloud computing, Web of Things, virtual containers

## Abstract

The data produced by sensors of IoT devices are becoming keystones for organizations to conduct critical decision-making processes. However, delivering information to these processes in real-time represents two challenges for the organizations: the first one is achieving a constant dataflow from IoT to the cloud and the second one is enabling decision-making processes to retrieve data from dataflows in real-time. This paper presents a cloud-based Web of Things method for creating digital twins of IoT devices (named *sentinels*).The novelty of the proposed approach is that sentinels create an abstract window for decision-making processes to: (a) find data (e.g., properties, events, and data from sensors of IoT devices) or (b) invoke functions (e.g., actions and tasks) from *physical devices* (*PD*), as well as from *virtual devices* (*VD*). In this approach, the applications and services of decision-making processes deal with sentinels instead of managing complex details associated with the *PDs*, *VDs*, and cloud computing infrastructures. A prototype based on the proposed method was implemented to conduct a case study based on a blockchain system for verifying contract violation in sensors used in product transportation logistics. The evaluation showed the effectiveness of sentinels enabling organizations to attain data from IoT sensors and the dataflows used by decision-making processes to convert these data into useful information.

## 1. Introduction

IoT devices are becoming a key element in decision-making processes [1,2,3]. These devices are quite common in multiple infrastructures, such as Industry 4.0 [4], healthcare domain [5], and supply chains [6], to name a few. The data produced by these devices follow a lifecycle from the sensors to the edge [7], to the fog [4], and to the cloud [8]. In this lifecycle, data are acquired (mainly at the edge [9]), prepared and analyzed (typically at the fog or the cloud [10]), and finally converted into information for human consumption to use it in decision-making processes (mainly at the cloud [8] through end-user devices). In these types of infrastructures (any combination of edge, fog, or cloud), the virtual containers (VC) are key to deploy services on each infrastructure [11,12,13]. These services provide dataflows from the IoT to the cloud that produce different types of data and information, which proves to be key for organizations to conduct critical decision-making processes [14,15,16].

However, extracting data and information from these dataflows to deliver it to decision-making processes in real-time represents a huge challenge in two directions: the first one is verifying the accomplishment of a constant dataflow from IoT to the cloud; and the second one is enabling decision-making processes to retrieve, in real-time, data and information from different points of dataflows. These data acquisition tasks through dataflows are not straightforward because of the heterogeneity of the components participating in a dataflow (applications, types of sensors, data formats, infrastructures [17], to name a few). It is desirable a manner not just to acquire data and information from dataflows, but also to invoke actions and tasks on the dataflow components. That could facilitate tasks on decision-making analysis.

We propose to create *digital twins* of the IoT data acquirers (hardware, physical machine, or virtual container, application, or microservice) by using Web of Things cards (WoT) (www.w3.org/WoT accessed on 5 August 2021) for decision-making process to retrieve, in real time, data and information or invoke actions or tasks. A *digital twin* is an abstract representation commonly used in Industry 4.0 for IoT device monitoring [18]; that is, a virtual replica of objects or processes that simulate the behavior of their real counterparts. WoT is an initiative for representing and managing definitions of IoT artifacts (devices, components, applications, etc.), which suggests using a set of well-accepted protocols from the Semantic Web for any IoT artifact from the physical world to be available into the World Wide Web by creating a net of WoT definitions [19].

In this paper, we present the design, implementation, and evaluation of a cloud-based WoT method for creating *digital sentinel twins* (DST) of IoT devices. A *DST* creates an abstract window for decision-making processes to attain information and data, such as properties, events, and data produced by sensors, and to invoke actions or tasks from IoT devices. An IoT device is a *physical device* (PD) with sensors and tasks that can be accessed directly or through a *virtual device* (VD). A VD is an application or microservice encapsulated into a virtual container for acquiring, extracting, processing, monitoring, and analyzing data from *PDs*. Figure 1 shows an example of the process used by a *DST* to create a window for decision-making processes consumption (by either a human, application, or *VD*). As it can be seen, in this approach, the applications or services used in decision-making processes deal with *DSTs* instead of managing the complex details associated with the *PDs*, *VDs*, or cloud computing infrastructures.

We implemented a prototype based on this method to perform case studies supported by GPS, temperature, and speed sensors. Additionally, using a blockchain system, the compliance of contracts to which these sensors are subject in the transportation logistics of products is continuously verified. The evaluation revealed the effectiveness of the *DSTs* for organizations to attain data and information about both IoT devices and the whole processes converting IoT data into useful information required in decision-making processes.

The contributions of this work are:The design, implementation, and evaluation of a cloud-based WoT method for creating digital sentinel twins of IoT devices.The definition of the digital sentinel twin concept as a mean for accessing data and information, and for invoking tasks from IoT devices.

The rest of the paper is organized as follows: Section 2 describes the state of the art of works related to the topics of the proposed method; Section 3 describes the design and construction of a method to create *DSTs* for interacting with IoT devices; Section 4 describes the implementation of a prototype for the creation of *DSTs*; Section 5 presents the results of the prototype in two phases of experiments; The discussion of the obtained results is described in Section 6. Finally, Section 7 is presented with conclusions and future work.

## 2. Related Work

In the literature, there are some works about digital twins that are relevant to our approach, and these are next described.

In the context of digital twins, there are different works focused on its use for simulation, monitoring, risk prevention, etc., for IoT devices. Some are [20,21,22]. In [20], Assad et al. proposed a web-based digital twin (WDT) architecture, with the purpose of improving the sustainability of industrial cyber-physical systems. In [21] Bevilacqua et al. proposed a digital twin reference model for risk prediction and prevention. The difference between our work and these two proposals is that we establish the use of virtual containers in a middle layer to access, acquire, extract, transform, etc., the information of the IoT devices; in this way, through a DST, we are able to represent both the physical (the IoT artifact) and virtual (software applications accessing the IoT artifact) device. In [22] Gao et al. proposed a method of simulation and modeling in real time for the production line of digital twins. The effectiveness of the proposed method is verified by taking an assembly line as an example.

In the context of digital twins using virtual containers for the acquisition of information from IoT devices, the proposals [23,24,25] are interesting. In [23] Alaasam et al. proposed a study on live stateful stream processing migration of digital twins. The authors emphasized the importance of two factors that influence the construction of stateful stream processing in systems as complex as digital twin: Stateful virtualization infrastructure and the stateful data. In [24], Tingyu et al. proposed a methodology of container virtualization based on simulation as a service (CVSimaaS), the authors use virtual containers to implement a digital twins system, obtaining a lower consumption of resources with high efficiency. Like our proposal, these two works include the concept of virtual containers together with digital twins for IoT devices. However, these two proposals do not add a standardized representation to the digital twin. Moreover, in our proposal, we follow the WoT guidelines for the creation of the DST as universal accessible entities. In [25] Borodulin et al. proposed a model for simulation and prediction of industrial processes using digital twins in digital twin-as-a-service (DTaaS), which is a way to implement an orchestration of a set of independent services and provide scalability for simulation.

In the context of virtual container modeling, two proposals stand out [26,27]. In [26], Paraiso et al. presented an approach to model-driven management of Docker containers, which enables verification of the virtual container system architecture at design time. In [27], Piraghaj et al. proposed a simulation architecture called *ContainerCloudSim*, which was used to evaluate resource management techniques in virtual containers from cloud environments. Unlike these proposals, whose focus is only on virtual containers modeling, our proposal additionally models the environment of the IoT devices, adding WoT recommendations for representing them, which produces a DST flexible for consumption of the virtual containers and IoT devices data. In [28], Medel et al. proposed a performance model for Kubernetes-based deployment using Docker containers. Such a model can be used as a basis to support resource management and application design.

In the context of the use of virtual containers for the monitoring, simulation, and orchestration of IoT devices, there are two proposals [29,30]. In [29], Alam et al. proposed a modular and scalable architecture for IoT based on lightweight virtualization. Thus, the modularity provided, combined with the orchestration provided by Docker, simplifies management and enables distributed deployments, creating a highly dynamic system. In [30], Muralidharan et al. proposed a distributed monitoring system based on virtual containers for IoT applications for the management of a smart city environment. They achieved low latency, reliable and secure communication between large-scale deployment of IoT devices, with a focus on horizontal interoperability between various IoT applications. Both works do not use the digital twin concept, unlike our work (DST), which allows us to create a reflection with the properties and characteristics of the IoT device.

Muralidharan et al., in [31], proposed a semantic digital twin model for interacting with IoT devices. The authors used virtual containers to mimic IoT devices. This is the most similar approach to our proposal. However, they only focus on modeling the physical devices (PD) and not virtual devices (VD). Instead, through the DST, we can represent both the physical and virtual devices.

## 3. On the Building of Digital Sentinel Twins for IoT Devices

A *digital sentinel twin* (*DST*) is a software object produced from a data structure named WoTcard, which is created from data of *physical devices* (PD) or *virtual devices* (VD) interacting with surrounding elements for accomplishing some task involving sensors.

The conceptualization of a *DST* is illustrated in Figure 2, which is composed of the concepts next described.

A PD represents an IoT device interacting with sensors. A VD represents the software components required for creating a dataflow from an IoT device to a decision-making process. This means that a VD comprises components such as *Virtual Containers* (vc) or a *Virtual Container System* (VCS). A vc is a mechanism for logical encapsulation of software applications that creates environment independent applications required to create a dataflow. A VCS represents a set of $vc_{i}$ built as a single solution (service) to perform a task into the dataflow. A *containerized application* (CApp) is in charge of interacting with IoT devices, and it is encapsulated into either vc or VCS.

Thus, a DST is a versatile object for interacting in an easy manner with the complex and detailed structure of PD or VD. This is due to the flexibility of the *WoTcard*, which fulfills the recommendations of the W3C (www.w3.org accessed on 5 August 2021).This information comes from a *dataflow entity* (DfE), which captures information of each internal component (any of CApp∈vc, $vc_{i}\in VCS$, or PD), as well as relationships of these components with the PD. The DfE is basically a data structure including information about the structure, behavior, and function of VD or PD. The structure, behavior, and function are used to model the dataflow from the IoT device to the decision-making processes (as it captures these features of all entities participating in such a dataflow). The context of generation and usage of a DST is illustrated in Figure 3.

We considered an additional layer for standardizing the representation of a DfE by using WoT guidelines; this produces a WoTcard. That means, a WoTcard represents the information of DfE through standardized concepts about virtual containers. These concepts come from an ontology based on the ISO norm ISO/IEC JTC 1/SC 38 (www.iso.org/committee/601355.html accessed on 5 August 2021). By following these WoT standards, a WoTcard can represent, in a well-defined manner and unique identity, a VD or PD, without any further adaption on DfE.

We propose a three-phase method to create a DST for a dataflow from IoT devices to decision-making processes. Figure 3 also shows the conceptual view of the stages of the methodology: *modeling* (phase 1), where the data of the VD is acquired and its elements modeled; then, in the *standardization* (phase 2), these elements are depicted into WoT cards, which are ready to be used in the *consumption* (phase 3). Next, each stage is described more in detail.

### 3.1. Phase 1: On the Usage of Functional Modeling for Building DfE

A VD or PD can be modeled as a process to achieve a goal. The *functional modeling* [32,33] is quite suitable for creating a representation of its structure, behavior, and function. This modeling has been used, over the past years, for successfully representing processes in multiple scenarios [34,35,36].

In the proposed method, all the dataflow participants are modeled as objects composed of low-level parts; the object has an objective, and its components contribute to achieve, together, such an objective by performing *tasks*, such as acquiring, manufacturing, preparing, or analyzing data produced by IoT sensors. The functional modeling is quite suitable for the IoT context where it is important not only to model the IoT devices but also the dataflow participants to describe the properties, events, and actions performed from the IoT devices to the decision-making processes (either at the fog or cloud). This approach also allowed us to model all the participants in the production of these dataflows (any of $vc_{i}$, VCS, or CApps), which, in fact, are having a behavior of chained processes. This model is captured into DfE, which describes the behavior (properties and events), function (tasks), and structure (interconnections) of all participants in the dataflow.

As a preparation step of this method, we assume the existence of a $vc_{i}$ (see VD in Figure 3) executing a transformation of data (*task*); independently of the number of internal $vc_{i}$ in a dataflow, these are modeled as one DST. Let us consider the simpler case, where one $vc_{i}$ is decomposed into its function, structure, and behavior, and stored in a DfE. This decomposition is represented by means of WoTcards, making the DfE as a DST ready for consumption. For the case of a VCS, occurs the same process by each individual $vc_{i}$, integrating individual functions as the overall function of the DST.

The objective of this phase is to obtain the three main modeling elements of a vc:*structure*, where the components of the vc and its relationships are specified;*behavior*, where the values of the attributes of components are specified, according to the function of the vc;*function*, where the main goal of the vc and the tasks required to achieve it are specified.

This phase starts by receiving the configuration file of a vc, in *YML* or *YAML* format. From this file all the data required to represent the vc are acquired.

Next, the main elements are described following a decomposition approach.

#### 3.1.1. Function

The function is the goal description of the vc. If the input file is of a VCS, the function is modeled as a composition of functions of the internal $vc_{i}$. The function makes reference to the task executed (*transformation*) on the dataflow. There are six base function for a vc:*source*, the capability to act as an infinite reservoir of data;*transport*, the capability to transfer data from one point to another, including from one medium to another;*barrier*, the capability to prevent the transfer data from one point to another, including from one medium to another;*storage*, the capability to accumulate data;*balance*, the capability to provide a balance between the total rates of incoming and outgoing dataflows;*sink*, the capability to act as an infinite drain of data.

Specialized functions can be derived from these base functions, such as *produce-data, acquire-data, integrate-data, consume*, to mention a few. All the functions may be connected to each other into flow paths or flow structures forming software structures.

Thus, each $vc_{i}$ has at least one application ($App_{j}$) performing some *transformation* ($tr_{k}$); defined as follows.
(1)$$VC=\{vc_{1},vc_{2},\ldots ,vc_{i}\}$$
(2)$$App=\{App_{1},App_{2},\ldots ,App_{j}\}$$
(3)$$Tr=\{tr_{1},tr_{2},\ldots ,tr_{k}\}$$
(4)$$\forall vc_{i}\in VC:vc_{i}⊃App_{j}$$
(5)$$\forall App_{j}\in App:App_{j}⊃tr_{k}$$

The $tr_{k}$ is the key element for representing the *function* of a $vc_{i}$.

A containerized application (CApp) represents one or a set of applications $App_{l}$, l<j, encapsulated into a $vc_{i}$.
(6)$$CApp=\{App_{1},App_{2},\ldots ,App_{l}\}$$

#### 3.1.2. Structure

The internal *structure* of a vc is commonly organized as software structures (e.g., patterns, pipelines, parallel schema, dataflow, etc.). The model of the vc must reflect this kind of organization. Thus, the *structure* of the vc is defined as a logical directed acyclic graph DAG, where nodes (*N*) represent the components ($comp_{i}$) that compose the vc, while the interconnections between nodes ($comp_{q}\to comp_{r}$) are established by edges (*E*), which are defined as follows.
(7)$$N=\{comp_{1},comp_{2},comp_{3},comp_{i}\}$$
(8)$$E=\{comp_{1}\to comp_{2},comp_{2}\to comp_{3},comp_{i-1}\to comp_{i}\}$$
(9)DAG={N,E}

The DAG is the key element for representing the *structure* of a $vc_{i}$.

#### 3.1.3. Behavior

The *behavior* of the vc is established by assigning values to its properties, that is, by associating the function of the vc with the infrastructure (*H*) defined in the configuration file. The $vc_{i}$ are deployed on H∈I, being *I* the whole infrastructure (e.g., a cloud). The consumption of a set of resources (*R*) of the specified infrastructure (processor—*CPU*, memory—*MEM*, file system—*FS*, and network—*NET*) is denoted as R∈H per each $vc_{i}$, which are observed by a set of metrics (*M*).
(10)R={CPU,MEM,FS,NET}
(11)$$M=\{total-usage,per-core-usage,\ldots ,m_{n-1},m_{n}\}$$

*H*, *R*, and *M* follow a hierarchy of elements defined as:(12)∀h∈H:h={r,r⊆R}
(13)∀r∈R:r⊃value,value∈R,m(value)

Equation (Equation 12) specifies that each physical computer *h* (where a $vc_{i}$ runs) has a subset of physical resources *r*. Likewise, Equation (Equation 13) specifies that each physical resource *r* has a value denoting the performance of *r* for $vc_{i}$, and a metric *m* observes that value for performance analysis.

Each resource *r* produces several values in the continuous numerical space. Thus, a huge set of values is generated per resource *r*. These values are used for computing *utilization factors* (*UF*), which inform about the status performance of a resource *r*. Although the resources produce a lot of values and data, we are interested in such values of UF that could initiate a *risk situation*. Then, according to the ISO 31000 standard (www.iso.org/iso-31000-risk-management.html) accessed on 5 August 2021 for risk management [37], the values of UF are discretized in scales: low∈[0,0.33), medium∈[0.33,0.66) and high∈[0.66,1]. These thresholds indicate the level of performance (*_lvl*) of each resource $r_{i}$, as indicates Equation (Equation 14).
(14)UF={CPU_lvl,MEM_lvl,FS_lvl,NET_lvl}

The UF of *CPU* in an instant of time *t* is defined by (Equation 15).
(15)$$U_{CPU}=1-\left[\frac{T_{CPU}-C_{CPU}}{T_{CPU}}\right]$$
where, $T_{CPU}$ is the total processing capacity of the physical computer, given by the sum of the capacity of each of the cores and $C_{CPU}$ is the *CPU* usage at the current time.

The UF of the file system in an instant of time *t* is calculated by (Equation 16).
(16)$$U_{FS}=1-\left[\sum_{i=1}^{f}\left(\frac{T_{FS_{i}}-C_{FS_{i}}}{T_{FS_{i}}}\right)\right]$$
where, *f* is the number of partitions available on the physical computer, $T_{FS_{i}}$ is the total capacity of the current partition on the physical computer, and $C_{FS_{i}}$ is the consumption of the current partition at a given moment. As shown, the multiple storage partitions associated to a studied object are considered in Equation (Equation 16).

The UF of memory is calculated by (Equation 17).
(17)$$U_{MEM}=1-\left[\frac{T_{MEM}-C_{MEM}}{T_{MEM}}\right]$$
where, $T_{MEM}$ is the total memory on the physical computer, and $C_{MEM}$ is the memory consumption at time *t*.

The UF of network is calculated by (Equation 18).
(18)$$U_{NET}=1-\left[\frac{T_{NET}-\left(TX_{NET}+RX_{NET}\right)}{T_{NET}}\right]$$
where, $T_{NET}$ is the total capacity of the network in bytes, $TX_{NET}$ is the number of bytes transmitted, and $RX_{NET}$ is the number of bytes received.

The set UF is the key element for representing the *behavior*.

As a result of this stage, a DfE is obtained, conformed by the three elements before described (structure, behavior, and function). The second stage of the method operates on this data structure.
(19)DfE={DAG,UF,Tr}

### 3.2. Phase 2: Standardized Access to DST by Means of WoT

At this point, a DfE provides a representation of the necessary data of the vc. However, we require a helpful representation to interact with the vc; such an interaction may be machine to machine or human to machine. For achieving this flexibility, this representation is based on the Web of Things (WoT) principles [38]. This standardized representation of a vc is named *WoT card*. In addition to the information captured by DfE, *metadata* of the vc are also added to the *WoT card*. These metadata are: IP addresses, volumes, ports, namespaces, etc. A *WoT card* is defined as shows Equation (Equation 20).
(20)WoTcard={DfE,metadata}

In the case of a VCS, such elements are defined recursively to capture data about structures and transformations used and performed by the whole VCS and its components, respectively.

According to the WoT recommendations, the generation of the WoT cards must be based on ontologies. In this sense, we defined and created an ontology (named *VC Docker FU Ontology* (available at github.com/adaptivez/VirtualContainerOntology accessed on 5 August 2021)), which can be adapted to the context of any WoT card in several scenarios. The *VC Docker FU Ontology* is used as a reference in the whole generation of WoT cards during the representation of vc. This ontology comes from two more ontologies, it extends from the *VC Docker Ontology* (github.com/langens-jonathan/docker-vocab/blob/master/docker.md#config accessed on 5 August 2021), which extends from the *VC ISO Ontology*. The latter ontology was created from scratch according to the norm ISO/IEC JTC 1/SC 38 (www.iso.org/committee/601355.html accessed on 5 August 2021), it defines all the concepts and constraints of the norm in an abstract manner. The *VC Docker Ontology*, in its original version, already defines concepts and constraints of virtual containers into the Docker environment, it was adapted in line with the *VC ISO Ontology*; some additional concepts and restrictions were included to fulfill with the ISO norm. The *VC Docker FU Ontology* adds concepts about the behavior related to infrastructure resources—*CPU*, *MEM*, *FS*, and *NET*—(such as levels of utilization and properties of such values), and function of virtual containers (such as base functions and tasks).

Technically a WoT card is based on an abstract class named *Thing*, which is the base object for modeling in the WoT approach. It is based on the representation structure of *Thing Description* (*TD*). *Thing Description* is the base model for describing any IoT Thing in the W3C Web of Things approach, it describes the metadata and interfaces of Things (www.w3.org/TR/wot-thing-description accessed on 5 August 2021). Thus, a WoT card is composed of three elements: (i) metadata (of Thing), which contains interactions (how Thing can be used); (ii) vocabulary, which contains concept definitions used into the *Thing Description* structure, useful for interactions; and (iii) URIs, which are useful to identify resources into *Thing Description*, these are Internet links denoting relationships between Thing and other resources on the WoT.

The WoT card was designed so that an external user can interact with it by asking about: *properties, actions*, and *events*. Properties contain information about the Thing, such as behavior (UF), structure (*nodes*, and *edges* of the DAG), and metadata of the VC. Actions refer to the functions of the Thing, including tasks (Tr*s*) executed by its components. Events refer to alerts on behavior changes, such as defined by the *utilization levels* (*CPU_lvl, MEM_lvl, FS_lvl, NET_lvl*).

Then, a WoT card is represented as a file following the format and structure of JSON-LD (*JavaScript Object Notation for Linked Data*, www.w3.org/TR/json-ld11 accessed on 5 August 2021). Listing 1 illustrates a portion of an example of WoT card.

**Listing 1.** Thing Description (TD) structure following the JSON-LD format.
 {

    "@context": "https://www.w3.org/2019/wot/td/v1",

    "id": "996ba6e...aec5f14",

    "@type": "Thing",

    "td:title": { "@value": "..." },

    "td:description": { "@value": "..." },

    "properties": {

        "ctv:metadata": { data{} },

        "ctv:structure": { data{} }

    },

    "actions": { "ctv:functions": {input{}, output{}} },

    "events": { "ctv:behavior": {} }

 }


### 3.3. Phase 3: Consumption

After the WoT card has been generated and its data stored, it is ready for consumption by means of a DST. For the DST to be accessible and consumed, it must become an intermediary between the modeled object (vc) and the consumer. This is possible by using a RESTful system, which can process requests with the most common HTTP actions: GET, POST, PUT, DELETE. In this way, any artifact making REST type requests can consume the DST. The consumption can be on properties, actions, or events, which are defined as follows.
(21)ConsumProperty={WoTcard,property}
(22)ConsumEvent={WoTcard,event}
(23)ConsumAction={WoTcard,action[input]}

Each element of the WoT card is universally identified and accepted by other physical or abstract entities (e.g., other vc, VCS, applications, devices, human-requests, etc.) by means of a Universal Resource Identifier (URI) in a unique and universal manner.

For the consumption of DST properties (Equation 21), it is necessary to give the URI of the DST and the specific property to access. Additionally, in the event consumption (Equation 22), the URI of the DST and the event to be accessed must be given. For invoking actions (Equation 23), it is necessary to give the URI of the DST, the action to be performed and the input required for that action as parameter. In the three types of consumption, a JSON object is obtained as a response indicating a *value* if a property or event were requested, or a *value* or *resultant flag* if an action was invoked. Next, an example of consumption of the property “platform” and the function “sum” are given.


*Request* (property):
https://www.example.com/wotmodel/containers/123456789/platform




*Response*:
{ "platform" = "Docker" }




*Request* (function):
https://www.example.com/wotmodel/containers/123456789/sum/2/3




*Response*:
{ "result" = "5" }



## 4. DST Prototype

This section describes the implementation of a prototype for building DSTs based on the proposed method. The components of this prototype and its interactions are depicted in Figure 4. The components were implemented as microservices (encapsulated into virtual containers), coded by using Python 3.0, except for the *observation* component, which was implemented by using JavaScript and PHP because of the nature of observation tasks. Next, each component is described.

The prototype was deployed on the Docker platform, but DSTs may be created from another platform, such as LXC (linuxcontainers.org/lxc accessed on 5 August 2021), Hyper-V (docs.microsoft.com/en-us/virtualization/hyper-v-on-windows accessed on 5 August 2021), or rkt (https://www.openshift.com/learn/topics/rkt accessed on 5 August 2021), where a vc can be represented by a *YML* or *YAML* file.

### 4.1. Representation

In this service, the configuration file (*YML*) of the VD is parsed to build the DfE, capturing structure, behavior, function, and metadata of the participants in a dataflow from an IoT device to the decision-making process. After the creation of DfE, the WoT cards are generated and its corresponding URIs defined. In this way, a decision-making process can consume the *WoT card* information (properties, events, and actions). The URIs must follow a defined *namespace*, as shows the Expression (Equation 24):(24)$$\begin{array}{c} \mathrm{http}://\mathrm{www}.\mathrm{example}.\mathrm{com}/\mathrm{wotmodel}/\mathrm{containers}/\\ \;\;\;\mathrm{container}\_\mathrm{id}/\{\mathrm{property},\mathrm{event},\mathrm{action}\} \end{array}$$

The WoT cards along with the DfE are stored in a MySQL database.

### 4.2. Listener

This service monitors the state (behavior) of a given VD (any of vc, VCS, or CApp). It is in charge of storing and keeping updated, in real-time, all the captured information by requesting status information from the Docker daemon and registering, in the database, each event producing a change on the VD. It also keeps a communication with the *supervision* service to reflect any change on VD utilization levels, which are also stored in the database.

### 4.3. Supervision

This service supervises the VD and performs the acquisition of metrics through an external agent, called *cAdvisor*. This is an API that provides information about the metrics of the vc and the physical computers on which it runs. When acquiring the values of the metrics, it calculates the behavior values of VD (utilization levels of resources used by VD). It also responds to requests from the *Listener*, which is monitoring the VD and returns values of utilization levels of resources (high, medium, low) about *CPU*, *MEM*, *FS*, or *NET*, as well as the timestamp when values were captured.

### 4.4. Observation

This service offers options for observing the VD (structure, behavior, and function). It is a web application with intuitive interface designed for human consumption. Three tasks can be performed: (1) Discovering of VDs, for searching the vc*s* or a specific CApp by using its properties (name, description, type, creator, owner, etc.); (2) Monitoring VDs, to know easily the behavior of the resources used by a VD by means of warning color signs (red for critical, yellow for intermediate, and green for normal) and its utilization level values; (3) Observing risk levels, to know the risk of failure of the applications by means of a graph denoting virtual containers in nodes and its relationships in edges.

### 4.5. Consumption

This service acts as a gateway, and is in charge of attending and processing requests from external users (human users, software applications, virtual containers, etc.) trying to consume or interact with the given VD. This is performed by using an API REST for GET, POST, PUT, and DELETE requests. Three types of consumption are considered: *properties, events, and actions* depending on the desired consumption/invocation. For properties and events, this service queries the WoTcard of the VD, then gets the corresponding data from the database to send it to the requester. For actions, the service queries the WoTcard of the VD, then establishes a connection to the corresponding VD, which executes the action and returns the result to the requester. All responses are into a JSON file. This is illustrated by invoking the clustering task kmeans with the parameters *k* and a dataset named data.


*Preparation*:
URI = https://www.example.com/wotmodel/containers/123456789/kmeans

input = {"k"=2,"data":[{"col1":1,"col2":0,"col3":2},

{"col1":2,"col2":1,"col3":1},{"col1":0,"col2":0,"col3":2}]}




*Request*:
request.post(URI,input)




*Response*:
{"result":

{"cluster1":[{"col1":1,"col2":0,"col3":2},{"col1":0,"col2":0,"col3":2}],

"cluster2":[{"col1":2,"col2":1,"col3":1}]}}



## 5. Results

The evaluation of the prototype for DST creation was conducted in two phases of experiments. In the first phase, the prototype was evaluated in a controlled manner to measure the response and service times in the construction of the DST and in its consumption. In the second phase, a case study is presented based on the creation of DST for a platform for continuous verification of contracts using a blockchain network.

Table 1 shows the infrastructure used by the VCS created for both cases of study.

### 5.1. Metrics

The performance of the prototype was evaluated by capturing the following metrics.

*Service time* (ST): The time required by a microservice (VD) to complete a given task.*Response time* (RT): The time observed by an end-user or a decision-making process to complete a given task. This time considers the initial time to attain data, create the representation, and store it in the database when an end-user builds a DST. This metric also measures the initial time when an end-user sends a request to the prototype and the time spent by DST to process it plus the time spent by it to deliver the results to the end-user.

### 5.2. Controlled Evaluation

To conduct the evaluation of the prototype, a containerized application (CApp) was deployed on the previously described infrastructure, one instance of the CApp running on one virtual container vc. This CApp extracts data from real traces produced by ECG medical devices (IoT devices for acquiring electrocardiogram (ECG) signals) [11], and builds workloads at a given rate time, following a synthetic distribution. An input parameter defines the amount of data to be included in the workload.

By using the CApp, several experiments were carried out by varying the number of vc and IoT data sources (ECG sensors), as well as the timing when the DST captures the behavior of the CApp; this latter is called slot.

We captured the ST and RT metrics for each experiment, each one was performed 31 times (according to the Central Limit Theorem [39]) to capture the median value of both ST and RT.

Different combinations of virtual containers (vc) and DST*s* (dst) were tested, these combinations were defined in the form vcW−dstZ, where *W* is the desired number of virtual containers (vc) in the combination, and *Z* is the desired number of DST*s*. That means 1 (of *Z*) DST watches *W* virtual containers. For example, Expression (Equation 25) means 1 DST watching 5 virtual containers, this results in a total vc=5. Expression (Equation 26) means 5 DST watching 5 virtual containers, this results in a total vc=25.
(25)vc5−dst1
(26)vc5−dst5

These combinations also was executed by varying the *slot* parameter from 1, 10, to 100 s. Each combination of these parameters produces a median value of ST and RT, which are evaluated to show the behavior of the DST costs. The total time of ECGs extraction was 10 min.

#### Analysis of Results

Figure 5 shows, in vertical axis, the ST and RT by two key operations related to the building of a DST (GetData and StoreData tasks) produced by the different number of virtual containers, evaluated in these experiments. This experiment only shows the ST and RT observed by either end-users or a decision-making application. As it can be seen, the prototype can build in just seconds DSTs for multiple VCS (17.5 s for creating DSTs for 100 applications, each connected to an IoT data source). This time is only spent by the prototype once, which means that this is affordable for many environments. Moreover, the GetData task (parsing YML files and creating the DfE), as it was expected, was the more significant task in the building of a DST, whereas StoreData task (indexing the DfE in a database) results were not significant for the DST building RT.

Figure 6 shows, in vertical axis, the ST (for the Representation task) spent by the building of the DSTs according to the sequences of DST*s* and virtual containers evaluated in these experiments (horizontal axis). As expected, the more the number of DST*s*, the more the ST spent by the prototype to create the representation of these DSTs.

Figure 7 shows, in vertical axis, the RT spent by the DSTs to retrieve information about VDs and PDs to the end-user (in this case a DST client application) per different sequences of DST and virtual containers (horizontal axis) for different time slot. It can be observed that increasing the number of vc*s* per DST also increases the number of requests performed by the DSTs per slot, increasing RT. The RT obtained is acceptable as soft real time [40].

### 5.3. A Case Study: Blockchain Network for Continuous Contract Verification

The previous evaluation showed the costs in time associated to create DSTs for decision-making process to attain IoT data (by using simple REST API) without dealing with technology elements from IoT and cloud, just invoking tasks on *DSTs*.

We also conducted a case study to show the flexibility of DSTs into a dataflow composed by an end-user (human, device, or application), DSTs, virtual containers (VDs), and IoT devices with sensors attached (PDs). This dataflow was emulated from a real trace of a logistic transportation of a supply chain of food, which is used by a VCS implementing a blockchain service for the verification of contract violations by monitoring GPS, temperature, and speed sensors of a set of transportation trucks [41].

Figure 8 shows the conceptual representation of this case study. As it can be seen, two DSTs were created for two VCS (including three virtual containers). The DSTs deliver to end-users or applications (decision-making processes) information about VDs (the system) and PDs (physical devices).

Figure 9 shows, in the horizontal axis, a timeline of the tasks performed by participants on the dataflow (vertical axis) of verifying contract violations: *Build* (*tsk1*), *data acquisition of temperature* (*tsk2*), *data acquisition of speed* (*tsk3*), *data acquisiton of GPS* (*tsk4*), *send request* (*tsk5*), *get data* (*tsk6*), and *deliver request* (*tsk7*). The timeline for this case study was 10 min. In *tsk1* the prototype builds two DSTs. Then, the data acquisition was carried out from IoT sensors (*tsk2*, *tsk3*, and *tsk4*) by the virtual containers, which were stored on the blockchain network. Additionally during the timeline, every 10 s, the virtual containers verified, registered, and reported contract violations on the blockchain network: first the consumer requests to DST (*tsk5*), then the blockchain is queried by the corresponding virtual container (*tsk6*), and finally the DST responses to the consumer (*tsk7*). As it can be seen, the impact of the DST creation (*tsk1*) and communications (*tsk5* and *tsk7*) is not significant in comparison with the time spent by get data from the blockchain network (*tsk6*) and the data transfer from sensors to the blockchain network (*tsk2*, *tsk3*, and *tsk4*). Figure 9 also shows that DST can capture the data produced by both, VD*s* (*tsk6*), and PD*s* (*tsk2*, *tsk3*, and *tsk4*).

We observed that DST*s* were able to inform to end-users, on demand and in real time, about contract violations. From the total number of requests (47) to the DST*s*, just in 3 requests the DST*s* informed contract violations.

The DST*s* can also deliver, on demand and in real time, the data rate produced and received by PD*s* to the end-user. Therefore, the behavior of the PD*s* can be known by end-users in decision-making time by analyzing these data. In this case study, the prototype showed a regular data production from sensors, with a reduction and increment of the data rate. This could imply to a potential bottleneck in the reception of data or a possible inconsistent data production from sensors at IoT devices. Figure 10 shows the received data amount of 47 user requests to the DST*s*.

The averages of consumed resources *r* (processor—*CPU*, memory—*MEM*, file system—*FS*, and network—*NET*) by the prototype in the case study are shown in Table 2. To obtain them, first the consumption of such resources were measured before and during the case study, this was carried out 32 times (w=32). Then the differences between initial ($r_{k_{ini}}$) and final ($r_{k_{fin}}$) values were computed and added. Finally the average of the differences were obtained, as shows Equation (Equation 27).
(27)$$r_{k_{avg}}=\frac{\sum_{i=1}^{w}\left(r_{k_{fin_{i}}}-r_{k_{ini_{i}}}\right)}{w}$$

It is important to note that the blockchain network is not of exclusive use of this prototype, it can be consumed by external applications. This VCS (blockchain network) can be replaced by other VCS (e.g., a data analytics system), in such a case that the DST must deliver the data produced by this new VCS without performing deep changes, but rebuilding the DfE of the DST.

## 6. Discussion

In this paper, we demonstrated the viability of the proposed method by applying the implemented prototype in two scenarios. The first one is regarding a controlled evaluation for extracting data from traces produced by ECG medical devices. This scenario showed the response and service time performance during the building and consumption of DST*s*. The second scenario demonstrated the flexibility of DST*s* to attain information (verification of contract violations on a blockchain network) in real-time from a dataflow of transportation logistics.

The obtained prototype was tested on distinct scenarios for intermediate and partial experiments before obtaining the results reported in this paper. In all these experiments, the prototype showed good performance in several tasks, such as discovering vc*s*, monitoring VCS, supervising CApps through created DST*s*. Several interactions were performed on these DST*s*, accessed by other CApp*s*, human requests, and software applications.

According to the results of the controlled evaluation (Section 5.2), we can see that augmenting the number of vc*s* per DST increases exponentially the response time for both the building and consumption of DST*s*. The building of 1 DST with 5 vc (vc5-dst1) takes an average response time of 0.90 s (see Figure 5). The consumption of the DST with the same configuration (vc5-dst1) takes an average response time of 0.52 s (see Figure 7).

The case study (Section 5.3) supports the results achieved in the controlled evaluation. In this case, the average response time during the building of the DST*s* (sequence vc3-*s*2) was 1.2 s (see Figure 9). For the consumption of the DSTs the average response time was 13 s (see Figure 9). However, it is important to note that from these 13 s, 10 s correspond to the communication to and from the blockchain networks for obtaining data. Thus, we can conclude that 3 s is the real response time for the consumption.

In all the experiments of the prototype, the interaction with the created DSTs was easy because complex requests were not necessary. The benefits of using the created DSTs are as follows:Standardized interaction. Since a WoT card is based on W3C guidelines, a DST can be consumed by distinct users (humans, devices, or applications);Easy consumption. Through a DST, users can: (a) access to data, properties, and events; and (b) invoke tasks and functions, both directly on target devices (VD*s* or PD*s*);Flexible access. A DST can be exploited by external users by means of RESTful requests from distinct locations to the one of the DST environment;Decision-making aid. DST*s* can be used as a mean in decision-making tasks (discovering, classification, monitoring, supervising, migration, to mention a few);Generation of DST. The building of DSTs is quite simple and transparent if a well-structured file configuration (*YML* or *YAML*) is given;Minimal required resources. The execution of DSTs requires minimal infrastructure resources (*CPU*, *MEM*, *FS*, and *NET*).

## 7. Conclusions

This paper presented a cloud-based WoT method for creating digital twins of IoT devices, named (*digital sentinel twins—DST*). A DST is an object that abstracts physical or virtual devices to operate over them by consuming its properties, events, or invoking its functions. This object has the advantage that by investing minimal time and resources, an external user (human, software application, or virtual devices) can access to all the data and functions of those devices. That is useful for interacting with IoT devices in several scenarios.

The method comprises three phases: (a) Modeling, where the data of the VD or PD are acquired, with these elements that device is modeled, generating a *dataflow entity* (DfE); (b) Standardization, where the elements of the model are represented into a standardized representation named *WoT card*; this representation follows the guidelines of the Web of Things to make its elements universally accessible by means of URIs; and (c) Consumption, the advance of the *WoT card* generated is that it can be consumed in external scenarios by distinct users (human, software applications, or virtual devices) in different ways.

Based on the proposed method, a functional prototype was implemented. This prototype was tested by creating DSTs in several experiments considering distinct scenarios of use (discovering and monitoring of VCs and applications, supervising CApps, etc.). By means of the created DSTs, it was possible to consume data and invoke functions of virtual and physical devices. In this paper, two experiments were reported to demonstrate the viability of the proposed method, creating flexible and useful DSTs. The first experiment was to show the spent time for creating and consuming DSTs. The second one was to demonstrate the use of DSTs into a scenario of a blockchain network for verifying contract violation on sensors used in product transportation logistics.

A DST creates an abstract window for decision-making processes to attain information and data from virtual and physical devices. It acts as a useful mechanism to interact with those devices in several scenarios. Its creation is not expensive in terms of time and computational resources, and it produces a access to data and functions of the target devices. These characteristics may be obtained without managing complex details associated to virtual and physical devices and cloud computing infrastructures.

Nevertheless the benefits obtained by the proposed method, it is important to mention some limitations of the proposed work:The creation of DST*s* only can be achieved if a well-structured configuration file is given, in *YML* or *YAML* format;A DST has no other way to consume it that RESTful requests;When target devices (VD*s* or PD*s*) and DST*s* reside in the same infrastructure, the response time of performed tasks increases exponentially.

As further work, the inclusion of security aspects into the *DSTs* is considered; this will enable its manageability and control while maintaining its flexibility of use.

## Figures and Tables

**Figure 1 sensors-21-05531-f001:**
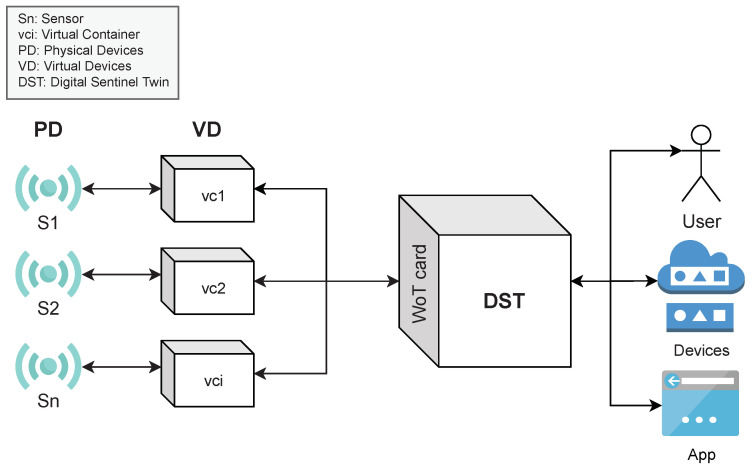
Conceptual view of a *DST*.

**Figure 2 sensors-21-05531-f002:**
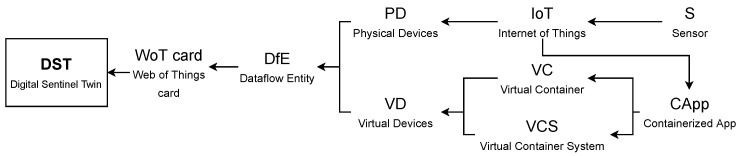
Conceptualization of a *DST*.

**Figure 3 sensors-21-05531-f003:**
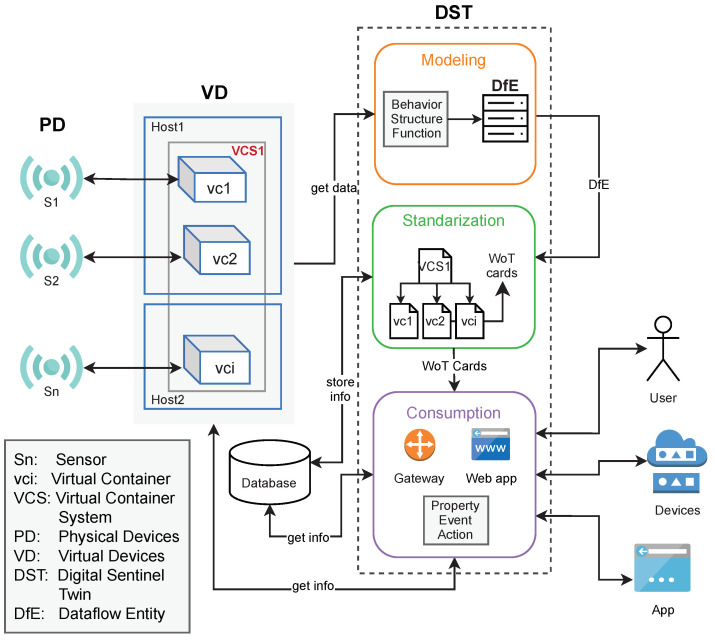
Context of a *DST*.

**Figure 4 sensors-21-05531-f004:**
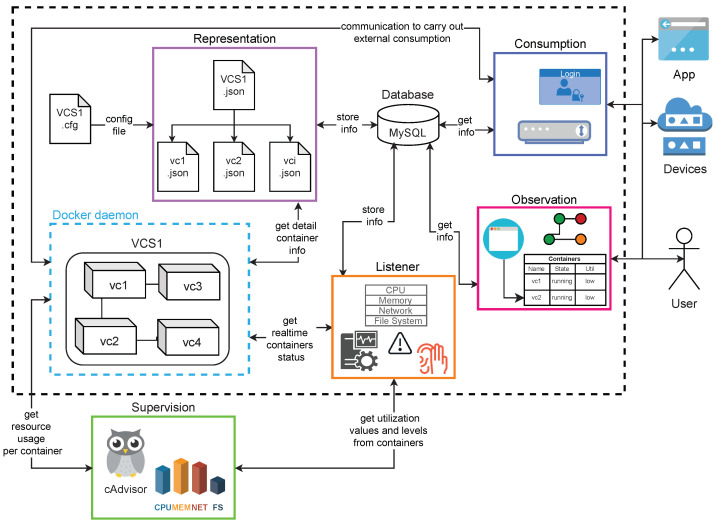
Components of *DST* prototype.

**Figure 5 sensors-21-05531-f005:**
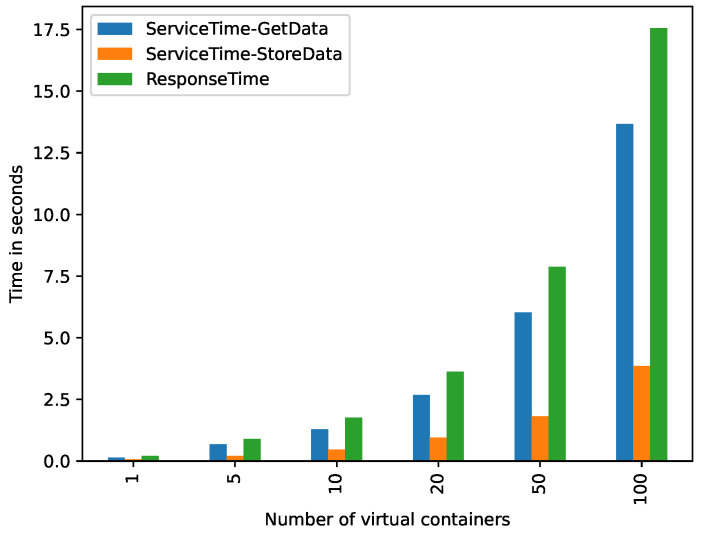
Service and response times produced by the tasks *GetData* and *StoreData*.

**Figure 6 sensors-21-05531-f006:**
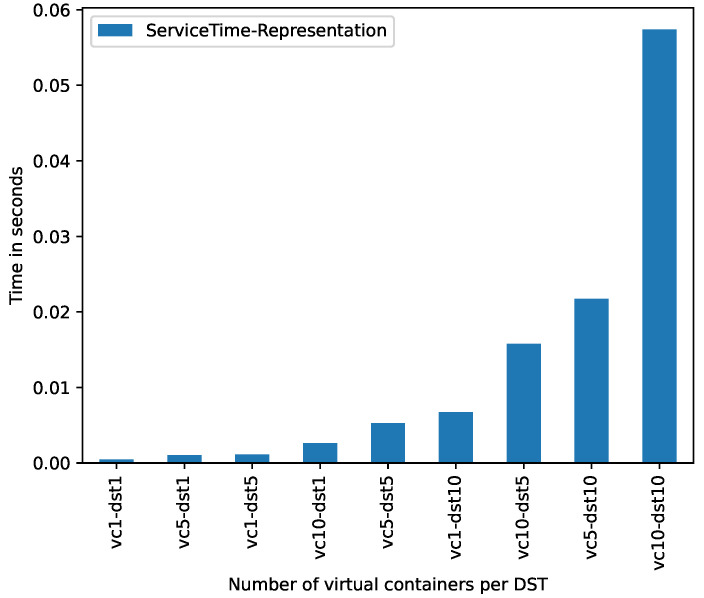
Service time produced by *representation* component.

**Figure 7 sensors-21-05531-f007:**
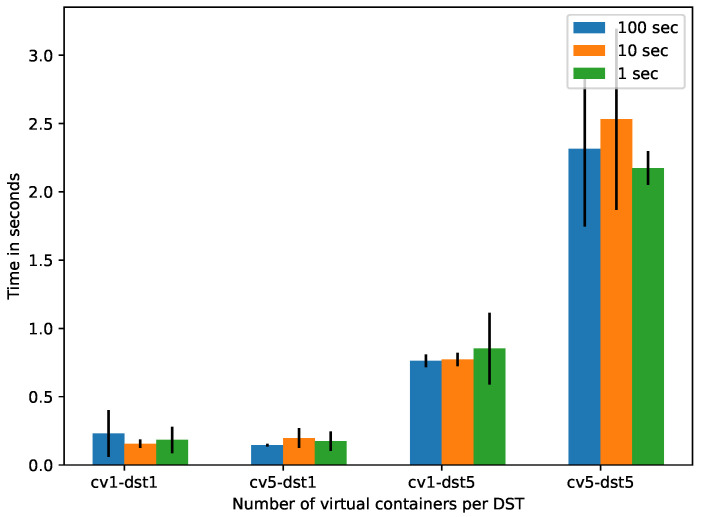
Response time in *DST* consumption.

**Figure 8 sensors-21-05531-f008:**
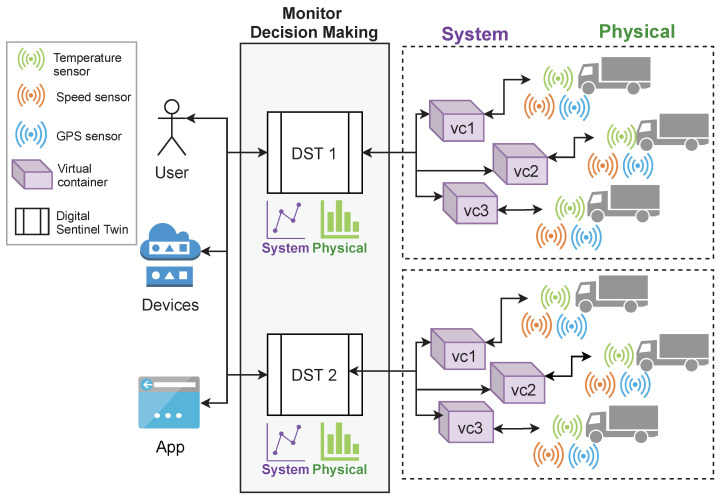
Conceptual representation of the scenario for the case study.

**Figure 9 sensors-21-05531-f009:**
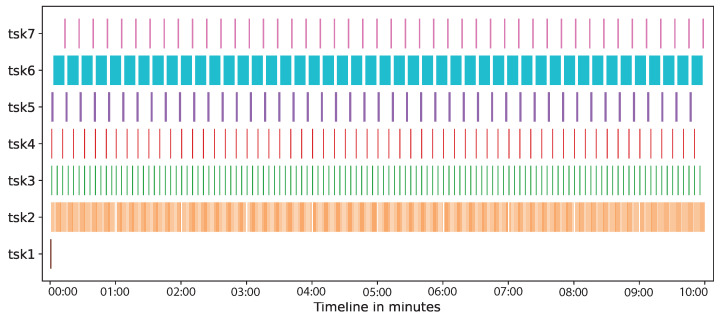
Time of tasks in the case study.

**Figure 10 sensors-21-05531-f010:**
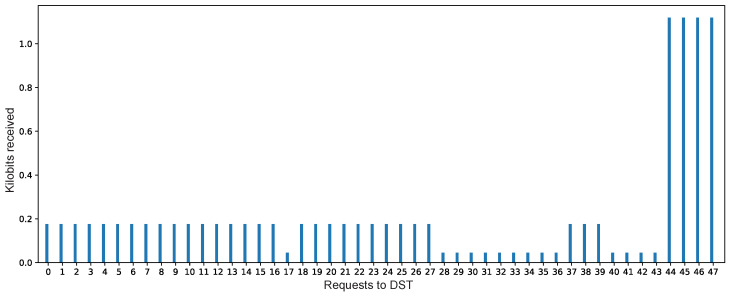
Kilobits received in the requests.

**Table 1 sensors-21-05531-t001:** IT Infrastructure used in the experiments.

ID	Cores	Processor	MEM	HDD	OS
Server1	4	Intel(R) Core i5	16 GB	256 GB	macOS BigSur
Server2	12	Intel(R) Xeon(R) E7-4830	128 GB	1 TB	CentOS 7

**Table 2 sensors-21-05531-t002:** Average values of resource consumption.

*CPU* (%)	*MEM* (Megabytes)	*FS* (Megabytes)	*NET* (Megabits/s)
2.306	33.884 (0.02%)	20.109	9.556

## Data Availability

Not applicable.

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
