# Peer review of "A WoT-Based Method for Creating Digital Sentinel Twins of IoT Devices"

_sensors, 2021, doi:10.3390/s21165531_

Round 1

Reviewer 1 Report

Conceptual Issues
=================

- this is another level of abstraction - question is whether that adds value
  - you define a structured way of storing and obtaining information
  - but the decision maker (application) is very domain specific
  - so, the decision maker must anyway specify what information required in the "window"
  - usually, the decision maker will directly interface with WoT
  - so, question -> is the overhead still beneficial?
- it appears, your evaluation is based on a DST enabled environment
  - but your comparison should have been of response times with/without DSTs
  - should also have a comparative resource usage
- I am not sure whether your set theory representation of relationships brings clarity
- protocols such as MQTT provide a number of facilities that enable applications to perform such tasks as you describe
  - so again, my concern with the usefulness of your architecture

Language Issues
===============

- the grammar or language related issues mentioned below are quite obvious even for a non-native speaker, please address them
- many grammar issues, some examples "...composed by...", "Structure, where are specified..."
- ways of describing are very confusing, some examples "...which is composed of the next concepts.", "...fashion and unique identity a..."

Author Response

Response to Reviewer 1 Comments

The authors would like to thank the reviewer for his/her overall positive assessment of our paper. The reviewer has clearly pointed out what we also consider are the main strengths and contributions of our research. We have carefully reviewed the manuscript as suggested by the reviewer. Next we answers the comments of the reviewer.

Major modifications on the manuscript are marked in blue color.

Point 1: - this is another level of abstraction - question is whether that adds value

- you define a structured way of storing and obtaining information

- but the decision maker (application) is very domain specific

- so, the decision maker must anyway specify what information required in the "window"

- usually, the decision maker will directly interface with WoT

- so, question -> is the overhead still beneficial?

Response: Yes, we consider our approach adds a useful level of abstraction to IoT physical (PD) and virtual (VD) devices. Our approach is not just like a message passing method, it is an interface for both, a) accessing to data of variables and b) for invoking functions on PD/VD. We assume sensors are attached to PD or accessible by a VD. Thus, a Digital Sentinel Twin (DST) is a wrapper that allows interact to the public data and functions that an IoT PV/VD offers; the user can not access to private data or functions.

Point 2: - it appears, your evaluation is based on a DST enabled environment

- but your comparison should have been of response times with/without DSTs

Response: The experimentation environment is a scenario designed initially for modeling supply chains on the project "New Data Intensive Computing Methods for High-End and Edge Computing Platforms (DECIDE)". Ref. PID2019-107858GB-I00 from the Spanish Ministry of Science and Innovation Project. A previous publication on this scenario is https://doi.org/10.1007/s10586-021-03252-0. For the current paper, two DSTs were attached to this scenario, and no more additions were made. From the DSTs, data and functions (of the "original" PD/VD) were requested and invoked respectively.

- should also have a comparative resource usage

Response: Done. We added a comparison of consumed resources for the case study (Table 2). The Equation (27) describes how these values were computed.

Point 3: - I am not sure whether your set theory representation of relationships brings clarity.

Response: Done. We corrected and adapted the formulas.

Point 4: - protocols such as MQTT provide a number of facilities that enable applications to perform such tasks as you describe

- so again, my concern with the usefulness of your architecture

Response: The MQTT is a protocol for accessing only data of variables on IoT PDs. Our approach is different because it allows a) the access to data of variables, and b) the invocation of functions/tasks on target devices (PD or VD).

Reviewer 2 Report

The paper seems to be interesting and rather original. It focuses on presenting a novel cloud-based WoT method for creating digital twins of IoT devices, so-called sentinels. However, the clarity of the presented text fails in several aspects. Although I do not feel qualified to judge the English language and style, I strongly feel that the authors must improve English substantially. Because of poor English, many places in the paper are not quite understandable. Some examples: 

  • Page 3, lines 108-112
  • Page 4, lines 141-143
  • Page 6, lines 196, 201 and 220
  • Page 7, lines 239-241 and 246-249
  • Page 16, lines 504-506
  • Page 17, lines 527-530, 538-539 and 547-548

to mention a few.

The second important aspect in which the paper fails is the clarity of mathematical and other notations and conclusions based on the mathematical deductions. Here are several related comments:

  • Page 5, line 171, "A VD or PD essentially is a process performing a task", but earlier (page 4) were both VD and PD defined as "A PD basically represents an IoT device interacting with sensors", and "A VD represents the software components ...". Clarify that, please!
  • Page 6, formulas (3) to (5), it is necessary to define what are App and tr with indices; for instance, mathematically, the formula (5) is not correct.
  • Page 7, formula (8), compare it with the formula (1)! The same right side is denoted as VC in formula (1) and N in formula (8). In formula (9),  the authors did not define a relation denoted by an arrow. Moreover, formula (14) needs some more explanation as it seems to be rather messy. And in formula (15), a couple of non-defined variables are used.
  • Page 8, line 269, DfE is called here a Dataflow structure, while earlier (Fig. 3), it is called a Dataflow Entry. Moreover, formula (21) uses metadata, but a definition is missing.
  • Page 13, expressions (26) and (27) need more explanations as their components are not defined formally.

OK, there are also positive features in the paper; for instance, there are many interesting figures of outstanding quality. And, of course, the selected references are prevailingly well related to the theme.

Nevertheless, it would be fine to improve the English used and take into account my comments listed above. The paper has a good potential to be published after all the improvements and corrections.

Author Response

Response to Reviewer 2 Comments

The authors would like to thank the reviewer for his/her overall positive assessment of our paper. The reviewer has clearly pointed out what we also consider are the main strengths and contributions of our research. We have carefully reviewed the manuscript as suggested by the reviewer. Next we answers the comments of the reviewer.

Major modifications on the manuscript are marked in blue color.

Point 1: The paper seems to be interesting and rather original. It focuses on presenting a novel cloud-based WoT method for creating digital twins of IoT devices, so-called sentinels. However, the clarity of the presented text fails in several aspects. Although I do not feel qualified to judge the English language and style, I strongly feel that the authors must improve English substantially. Because of poor English, many places in the paper are not quite understandable. Some examples:

Page 3, lines 108-112

Page 4, lines 141-143

Page 6, lines 196, 201 and 220

Page 7, lines 239-241 and 246-249

Page 16, lines 504-506

Page 17, lines 527-530, 538-539 and 547-548

to mention a few.

Response: Done. We corrected and adapted the indicated statements.

Point 2: The second important aspect in which the paper fails is the clarity of mathematical and other notations and conclusions based on the mathematical deductions. Here are several related comments:

Page 5, line 171, "A VD or PD essentially is a process performing a task", but earlier (page 4) were both VD and PD defined as "A PD basically represents an IoT device interacting with sensors", and "A VD represents the software components ...". Clarify that, please!

Response: Done. We corrected and adapted the indicated sentences.

Page 6, formulas (3) to (5), it is necessary to define what are App and tr with indices; for instance, mathematically, the formula (5) is not correct.

Response: Done. We corrected and adapted the formulas.

Page 7, formula (8), compare it with the formula (1)! The same right side is denoted as VC in formula (1) and N in formula (8). In formula (9), the authors did not define a relation denoted by an arrow. Moreover, formula (14) needs some more explanation as it seems to be rather messy. And in formula (15), a couple of non-defined variables are used.

Response: Done. We corrected and adapted the formulas. Formula (15) refers to the resources of formula (11): CPU, MEM, FS, NET. We homologated the nomenclature.

Page 8, line 269, DfE is called here a Dataflow structure, while earlier (Fig. 3), it is called a Dataflow Entry.

Response: Done. We adapted how to refer to DfE.

Moreover, formula (21) uses metadata, but a definition is missing.

Response: Done. We adapted the definition of metada:

"In addition to the information captured by DfE, metadata of the vc are also added to the WoT card. These metadata are: IP addresses,volumes, ports, namespaces, etc."

Page 13, expressions (26) and (27) need more explanations as their components are not defined formally.

Response: Done. We adapted and re-structured the sentences to clarify such issues.

OK, there are also positive features in the paper; for instance, there are many interesting figures of outstanding quality. And, of course, the selected references are prevailingly well related to the theme.

Response: Thank you for your comments.

Reviewer 3 Report

This paper presents a cloud-based WoT method for creating digital twins of IoT devices (sentinels). These sentinels create an abstract window for decision making processes to get information/data from physical (PD) and virtual devices (VD). In this approach, the applications/services of the decision-making processes deal with digital sentinels instead of managing complex details associated to the PDs, VDs and cloud computing infrastructures. A prototype based on the proposed method was implemented to conduct a case study based on a blockchain system for verifying contract violation in sensors used in product transportation logistics. The evaluation showed the effectiveness of digital sentinels enabling organizations to get data from IoT sensors and the dataflows used by decision-making processes to convert these data into useful information.

The paper is interesting, but it could be improved if the following changes are made:

  1. In the abstract, the authors should clearly state the novelty of this research.
  2. In the introduction, the motivation and contribution of this paper should be described.
  3. The quality of images should be improved.
  4. More bibliography about decision making even in different fields would be interesting. As such, the authors are advised to include the following references to enrich the content of the paper:
    • Troussas C, Krouska A, Sgouropoulou C. Improving Learner-Computer Interaction through Intelligent Learning Material Delivery Using Instructional Design Modeling. Entropy. 2021; 23(6):668. https://doi.org/10.3390/e23060668
    • Y. Wang, P. Li, Y. Tian, J. Ren and J. Li, "A Shared Decision-Making System for Diabetes Medication Choice Utilizing Electronic Health Record Data," in IEEE Journal of Biomedical and Health Informatics, vol. 21, no. 5, pp. 1280-1287, Sept. 2017.
  5. Please be sure to support several assertions (e.g. pseudocode, maths, etc).
  6. Section 5 should be better organized.
  7. Section 6 should be enriched.
  8. In section 7, the limitations are missing.

Concluding, it is a work that I enjoyed reading it. Congratulations to the authors.

Author Response

Response to Reviewer 3 Comments

The authors would like to thank the reviewer for his/her overall positive assessment of our paper. The reviewer has clearly pointed out what we also consider are the main strengths and contributions of our research. We have carefully reviewed the manuscript as suggested by the reviewer. Next we answers the comments of the reviewer.

Major modifications on the manuscript are marked in blue color.

Point 1: In the abstract, the authors should clearly state the novelty of this research.

Response: Done. We adapted the content of the abstract.

Point 2: In the introduction, the motivation and contribution of this paper should be described.

Response: Done. We adapted the introduction according to this comment.

Point 3: The quality of images should be improved.

Response: Done. We modified the images for better visualization. For an easy read, the old Figure 10 now is described by means of text.

Point 4: More bibliography about decision making even in different fields would be interesting. As such, the authors are advised to include the following references to enrich the content of the paper:

Troussas C, Krouska A, Sgouropoulou C. Improving Learner-Computer Interaction through Intelligent Learning Material Delivery Using Instructional Design Modeling. Entropy. 2021; 23(6):668. https://doi.org/10.3390/e23060668

Y. Wang, P. Li, Y. Tian, J. Ren and J. Li, "A Shared Decision-Making System for Diabetes Medication Choice Utilizing Electronic Health Record Data," in IEEE Journal of Biomedical and Health Informatics, vol. 21, no. 5, pp. 1280-1287, Sept. 2017.

Response: Done. We added references ([2] and [3]) to these works at the beginning of the Introduction.

Point 5: Please be sure to support several assertions (e.g. pseudocode, maths, etc).

Response: Done. We modified and adapted formulas and expressions in the whole manuscript. Some definitions were modified, and others were added to clarify ideas.

Point 6: Section 5 should be better organized.

Response: Done. We modified the text for a better organization of ideas. Some remarks were made for a better understanding.

Point 7: Section 6 should be enriched.

Response: Done. We modified the text for a better organization of ideas. The benefits of the proposed method were reorganiced and some remarks were included.

Point 8: In section 7, the limitations are missing.

Response: Done. We included the limitations of the proposed work.

Round 2

Reviewer 1 Report

- some potential grammar issues I saw
"It is desirable a manner... "
"...as a mean..."

Reviewer 2 Report

I was reading the corrected version of the paper rather carefully, one my comment after the other, and I have a relatively good impression from the corrections made by the authors. Although there still are some peculiar English formulations, the paper in the present form could be published. Maybe, it would be beneficial for the final version to correct all the English by a native English speaking person. From a scientific point of view, I found the paper to be already much readable and sound. The authors accepted the majority of my suggestions and corrected all the mistakes that I have pointed out. I am satisfied with the resulting version of the paper. I have no new comments.

Reviewer 3 Report

The paper has been improved based on the comments.